# The Role of Vitamin D as a Prognostic Marker in Papillary Thyroid Cancer

**DOI:** 10.3390/cancers13143516

**Published:** 2021-07-14

**Authors:** Ashank Bains, Taha Mur, Nathan Wallace, Jacob Pieter Noordzij

**Affiliations:** 1Department of Otolaryngology—Head and Neck Surgery, Boston University School of Medicine, Boston, MA 02118, USA; bains@bu.edu (A.B.); wallacen@bu.edu (N.W.); 2Department of Otolaryngology—Head and Neck Surgery, Boston Medical Center, Boston, MA 02118, USA; Taha.Mur@bmc.org

**Keywords:** vitamin D, papillary thyroid carcinoma, staging, prognosis

## Abstract

**Simple Summary:**

Thyroid cancer is the most common endocrine malignancy in the United States and papillary thyroid cancer is by far the most common subtype. Vitamin D has been postulated as a key modulator in several cancer-related pathways, although its contributions to thyroid cancer remain controversial. In this paper, we review the metabolic pathways of vitamin D and explore potential links to cancer-related mechanisms. In addition, we also explore the medical literature related to vitamin D as a prognostic marker for staging in papillary thyroid cancer.

**Abstract:**

The role of vitamin D in modulating several cancer-related pathways has received an increasing amount of attention in the past several years. Previous literature has found an abundance of evidence of vitamin D exerting an anti-proliferative, anti-inflammatory, and pro-differentiation effect in various types of cancers including breast, colon, prostate, and pancreatic cancer. Although the link between vitamin D and thyroid cancer remains controversial, both biochemical evidence and clinical studies have attempted to establish a link between papillary thyroid carcinoma (PTC) and vitamin D status. Furthermore, the use of vitamin D as a prognostic marker has received increased attention, both in regards to clinical outcomes and cancer staging. In this review, we briefly discuss the metabolism and proposed mechanism of action of vitamin D in the context of PTC, and explore links between modulators in the vitamin D pathway and progression of PTC. We provide evidence from both clinical studies as well as molecular studies of metabolic targets, including vitamin D receptor and activating enzymes exerting an effect on PTC tissue, which indicate that vitamin D may play a significant prognostic role in PTC.

## 1. Introduction

Thyroid cancer is the most common endocrine malignancy in the United States, accounting for 3.2% of all new cancer diagnoses [1,2]. It is estimated that nearly 45,000 new cases of thyroid cancer will be diagnosed in 2021 [3]. The most common subtype, papillary thyroid cancer, accounts for 80% of these cases and, assuming negligible change in mortality since 2018, will result in approximately 360 deaths [2,4]. 

Over the past 30 years, the incidence of thyroid cancer has increased dramatically, rising from 5.5 per 100,000 individuals in 1990 to 13.7 per 100,000 individuals in 2017 with an estimated 850,000 individuals living with thyroid cancer in 2017 [5]. Based on these trends, researchers have projected that thyroid cancer may become the fourth most common cancer in the United States within the next 10 years [6].

In contrast, the death rate for thyroid cancer has remained relatively stable at 0.5 per 100,000 persons [5]. The discrepancy between the rising increasing incidence and stable mortality may possibly be due to the increased screening and more widespread use of imaging modalities by clinicians, specifically to ultrasound [7]. The estimated 5-year survival of patients with thyroid cancer remains excellent, at 98.3% [5], with established prognostic factors including age, sex, tumor histopathology, tumor size, lymphovascular invasion and extra-capsular spread, nodal spread, TERTp mutation, and distant metastases [8,9]. It is important to note that this promising prognosis is not reflective of all histologic variants and is likely driven by the papillary sub-group, with a 10-year cause-specific survival of 99% [4].

In general, thyroid cancer can be categorized into four major types: Well differentiated, poorly differentiated, medullary, and anaplastic. Differentiated thyroid cancer comprises nearly 90% of all cases and can be further subcategorized based on histology: Papillary, follicular, and Hürthle cell [10]. Papillary thyroid carcinoma (PTC) is by far the most common subtype. It is more commonly encountered in women with a peak incidence during the fourth and fifth decade of life and the BRAF V600E mutation has been implicated as a major biomarker, being present in roughly 45% of cases [11].

Recently, academic interests have been directed towards establishing prognostic features to expedite the workup of cases more likely to be malignant and aggressive. In particular, vitamin D has emerged as a potential target for prognostic evaluation. Previous research has linked vitamin D levels as a prognostic factor in a host of other cancers, including breast, colon, and prostate [12,13,14]. In a meta-analysis of 14 articles, Zhao et al. [15] found a significant association between vitamin D deficiency and thyroid cancer, noting that vitamin D deficiency could potentially increase the risk of thyroid cancer by 30% compared to healthy controls. 

As cancer stage is an established indicator for long-term survival [16,17], we sought to review recent literature on vitamin D levels as they relate to staging and prognosis in PTC. Herein, we also review basic metabolic pathways of vitamin D, as molecular studies have found that several metabolites and modulators of the vitamin D pathway may also play a role in thyroid cancer progression. This was accomplished via The National Institute of Health’s PubMed database specifying for articles including “vitamin D” and “papillary thyroid carcinoma” or “stage of papillary thyroid carcinoma”. Further expansion of the search criteria to further investigate the implications of molecular markers and enzymatic interactions, such as BRAF V600E and CYP24A1, occurred to further elaborate on these interactions. All study types (clinical, translational, experimental) were considered and analyzed. While there were no date or language restrictions in the search criteria, all the articles were published in English after 2006.

## 2. Vitamin D

### 2.1. Metabolism

The primary source of vitamin D in humans is sunlight, with lesser contributions from diet and supplementations [18]. Ultraviolet exposure of approximately 290–315 nm wavelength causes a photolysis reaction in the stratum basale of the skin, converting 7-dehydrocholesterol to the previtamin D3, which is subsequently isomerized to vitamin D3 [19]. Dietary sources often provide vitamin D2, originating from ergosterol of yeast [18]. In the bloodstream, vitamin D binding protein (DBP) carries synthesized vitamin D3 or ingested vitamin D_2_ to target organs. Vitamin D delivered to the liver is enzymatically converted to 25-hydroxyvitamin D (25OHD), which is then carried to the kidneys by DBP. Finally, vitamin D is converted into its biologically active form, 1α,25-dihydroxyvitamin D (1α25OH2D), in the kidneys via a hydroxylation reaction by the enzyme 25(OH)D-1α-hydroxylase (CYP27B1) [20]. In contrast, hydroxylation of either 25OHD or 1α25OH2D at carbon 24, carried out by 25(OH)D-24-hydroxylase (CYP24A1), begins the inactivation pathway for vitamin D. Alterations to any step along this pathway, most notably CYP24A1 activity, may have significant effects on vitamin D action [21]. 

### 2.2. Function

Vitamin D receptor (VDR) was first identified in 1979 in the nucleus of cells in a variety of different tissues. Binding of the active form of vitamin D, 1α25OH2D, to VDR causes a nuclear complex to form and exert regulatory effects on gene transcription for a variety of end-targets through binding to the vitamin D response element (VDRE) in promoter regions of various genes [22]. Although traditionally vitamin D is known for its role in bone metabolism and calcium homeostasis, research has shown VDR activation to exhibit anti-proliferative, anti-inflammatory, and pro-apoptotic activity [23,24]. In terms of its effect on cancers, past literature has shown vitamin D to inhibit growth on breast, colon, prostate, and pancreatic cancer cells [25,26,27,28].

## 3. Vitamin D and Papillary Thyroid Carcinoma

While the role of vitamin D levels in various cancers has been thoroughly established in the literature, the specific role of vitamin D in PTC remains controversial. While multiple studies have found significant associations between vitamin D deficiency and increased thyroid cancer risk, those with a larger sample size have also found no relationship whatsoever. Similarly, some have sought to investigate the role of vitamin D as a prognostic indicator for staging and advanced disease. Therefore, the association between serum vitamin D and papillary thyroid cancer remains elusive (Table 1). 

In their meta-analysis of fourteen studies, Zhao et al. [15] found a significant association between vitamin D deficiency and an increased risk of thyroid cancer, concluding that vitamin D deficiency may increase the risk for thyroid cancer by nearly 30%. A similar study of 276 patients by Hu et al. [29] found high serum vitamin D levels to be negatively correlated with thyroid cancer risk. More specific to PTC, Abdellateif et al. [30] studied 70 patients with PTC, 60 patients with benign nodules, and 60 healthy controls. They concluded that patients with PTC had significantly lower serum vitamin D and estimated a sensitivity and specificity of 65.7 and 58.3%, respectively, for vitamin D in the assessment of thyroid cancer. While the significant connection between vitamin D and PTC is reassuring, these findings could reflect the poor ability of vitamin D to predict or screen for PTC given the low sensitivity and specificity observed. This could help describe findings where evidence refuted any connection between PTC and vitamin D levels. In a Korean population study involving 5186 patients, Choi et al. [31] found that vitamin D levels could not predict thyroid malignancy in otherwise euthyroid patients. A study by Kim [32] had similar findings: Serum vitamin D levels could not predict malignancy in 410 patients with thyroid nodules. Overall, the relationship between vitamin D deficiency and PTC remains mixed.

The prognostic significance of vitamin D in PTC staging is also controversial. In their retrospective review of 334 patients, Sulibhavi et al. [33] found no significant relationship between cancer stage and serum vitamin D levels. However, patients’ meeting criteria for vitamin D deficiency were significantly more likely to have advanced disease. A similar study by Kim et al. [34] found a negative prognostic value in preoperative vitamin D levels, with low serum vitamin D being significantly associated with increased PTC staging. Furthermore, Roskies et al. [35] and Stepien et al. [36] both found lower vitamin D levels in PTC patients, with the latter study also finding a significant relationship between low serum vitamin D and advanced PTC staging. Sahin et al. [37] and Kim et al. [34] both had significant findings related to tumor size: Tumor diameter in PTC patients were significantly associated with serum vitamin D. These findings illustrate a possible connection between pre-operative vitamin D levels and prognosis in PTC.

One area of intrigue is the role of vitamin D in metastatic papillary thyroid cancer, particularly bone metastases. Clinckspoor et al. [38] and Yavropoulou et al. [39] identified differential expression of the vitamin D receptor and vitamin D enzymatic regulators in those PTCs prone to metastasis. Specifically, the tissue from lymph node metastasis had decreased expression of VDR when compared to the primary tumor. This cellular study suggests a strong local antitumor effect of vitamin D while also potentially linking the metastatic potential of PTC to the downregulation of vitamin D. 

While the aforementioned literature supports the connection between PTC staging and vitamin D, several studies have found no evidence of vitamin D as a prognostic factor in PTC. Warakomski et al. [40] found no significant relationship between either tumor size or clinical stage and serum vitamin D or vitamin D deficiency in 177 patients with PTC. Similarly, Ahn et al. [41] also found no correlation between vitamin D quartiles and both staging or prognosis in 820 PTC patients. Danilovic et al. [42] performed a retrospective review of 433 patients with thyroid nodules, finding no differences in vitamin D levels between malignant and benign cases and no effects on prognosis, agreeing with Kim [32] who also found no correlation with the occurrence of malignancy. 

As surgery is the primary treatment modality for PTC, the implications of treatment on vitamin D levels must be considered. Bone metabolism is one process which is heavily impacted by thyroidectomy and has potential implications on serum vitamin D. Hypoparathyroidism is common following thyroidectomy, with studies estimating anywhere from 19–38% incidence of transient injury and 3% of permanent deficiency following operative injury [43]. When studying post-menopausal women who underwent total thyroidectomy for differentiated thyroid cancer, those who underwent TSH suppression therapy with post-surgical hypoparathyroidism exhibited increased bone mineral density and decreased deterioration of bone microarchitecture when compared to their baseline parathyroid counterparts [43]. While some have proposed that pre-operative vitamin D levels play a role in predicting post-operative hypoparathyroidism, a recent study by Soáres et al. found no relationship in a small prospective study [44]. Therefore, it seems that vitamin D levels are not effective at predicting post-surgical parathyroid complications in those with thyroid cancer. 

## 4. Other Metabolites

Although the clinical value of vitamin D in the prognosis of PTC is controversial, several molecular studies have supported the involvement of vitamin D and its metabolites in the progression of PTC. Various targets in the pathway of vitamin D metabolism have been implicated, supporting the role of vitamin D in thyroid cancer. 

The inactivation of vitamin D begins with hydroxylation carried out by 25(OH)D-24-hydroxylase (CYP24A1)—in a gene expression analysis by Balla et al. [45] of 100 Hungarian thyroid samples from patients with PTC. CYP24A1 activity was positively correlated with tumor size, lymph node metastasis, and vascular invasion. Penna-Martinez et al. [46] studied 253 patients with either papillary thyroid cancer or follicular thyroid cancer versus healthy controls, finding that patients with vitamin D deficiency and certain haplotypes of the CYP24A1 gene had a higher risk of thyroid cancer. It is possible that alterations to the tumor microenvironment may lead to increased CYP24A1 activity in malignant tissue, creating a local deficiency in vitamin D levels and negating its anti-proliferative effects.

Another target in the vitamin D pathway implicated in thyroid cancer progression is the vitamin D receptor. Overexpression of VDR may affect the kinetics of vitamin D binding, thereby downregulating the effects of the protein complex on VDRE and effectively altering the downstream transcription of various proteins involved in tumorigenesis. Yavropoulou et al. [39] studied 45 thyroid samples from patients with PTC, finding both higher VDR and CYP24A1 expression in samples of malignant tissue. In addition, increased CYP24A1 expression was associated with lymph node metastasis. This may indicate that in addition to serum vitamin D levels, different haplotypes of CYP24A1 and VDR may serve as biomarkers of prognostic significance in PTC. Several other papers have found evidence of increased VDR or CYP24A1 expression in PTC tissue [38,47], as well as the correlation between increased VDR expression on worsening prognosis [48,49]. In addition, a cell culture study by Pang et al. [50] recently found that the knockdown of VDR attenuated the anti-proliferative effects of vitamin D in PTC tissue. Zhang et al. [51] found the increased expression of VDR to be significantly associated with PTC and that vitamin D induced apoptosis in PTC cell lines in a dose-dependent fashion. These molecular studies provide evidence of a more complex relationship between PTC tumor progression and vitamin D deficiency status. A more robust understanding of how the metabolic pathways of vitamin D may be differentially affected in patients with PTC may help physicians better delineate the role of the vitamin as a prognostic marker.

## 5. Conclusions

Molecular studies have established clear connections between several targets in the vitamin D metabolism pathway and thyroid cancer, including CYP24A1 and VDR. Despite these plausible biochemical relationships relating vitamin D to papillary thyroid cancer, the research regarding the use of vitamin D as a prognostic factor and the relationship with respect to vitamin D levels remains controversial. Further research is needed to further delineate these relationships and apply these biomarkers in the clinical care of patients with PTC.

## Figures and Tables

**Table 1 cancers-13-03516-t001:** Summary of literature focusing on vitamin D and papillary thyroid carcinoma.

Study Design	Study Author	Sample Size	Population	Results	Vitamin D Effect on PTC Stage
Clinical studies in humans					
Case series	Khadzkou et al. (2006)	35	Patients from Germany or Sweden with varying stages of PTC.	High VDR expression may be associated with greater tumor differentiation, thus potentially indicating a more favorable prognosis.	Elevated VDR/1α-hydroxylase activity was not associated with the stage of disease.
Case series	Clinckspoor et al. (2012)	136	Tissue samples gathered from patients undergoing thyroid surgery at two German hospitals.	Elevated VDR, decreased CYP24A1, and normal CYP27B1 levels were found when comparing malignant samples to benign entities. Low VDR expression in lymph node metastases when compared to the primary tumor; in agreement with Khadzkou et al.	N/A
Case series	Balla et al. (2015)	100	Hungarian, Caucasian patients with PTC.	CYP24A1 mRNA gene expression was increased in more than half of the samples when comparing PTC tissue to non-cancerous tissue in the same patient.	Increased CYP24A1 gene expression was positively correlated to several PTC staging variables including vascular invasion, LNM, hypothyreosis, and tumor size.
Case series	Yavropoulou et al. (2017)	45	Patients with PTC.	PTC is associated with altered expression of VDR and CYP24A1.	CYP24A1 expression was significantly upregulated and associated with PTC predisposition to vascular invasion, ETE, and LNM.
Case series	Zhang et al. (2018)	158	Chinese patients with thyroid nodules undergoing thyroidectomy.	Serum 1α,25-hydroxyvitamin D levels were higher in those with PTC as opposed to thyroid nodules. 1α,25-hydroxyvitamin D was shown to exhibit its downstream effects (proapoptotic and antiproliferative) via cAMP signaling.	Findings were statistically insignificant.
Case- control	Stepien et al. (2010)	76	50 with TC (27 PTCs, 16 FTCs, and 7 ATC) and 26 healthy controls.	Significantly lower 1α,25-hydroxyvitamin D concentration across all TC subgroups. No significant differences with respect to 25-hydroxyvitamin D concentrations.	Significant and progressive decreases in the circulating level of 1α,25-hydroxyvitamin D was noted in UICC stage I, II, III, and IVa.
Case- control	Penna-Martinez et al. (2012)	555	Patients at a large university German hospital (253 with TC and 302 healthycontrols).	While genotypic assays showed no differences, the expression of CYP24A1 haplotypes varies significantly among the two groups (TC and healthy controls).	N/A
Case- control	Sahin et al. (2013)	460	Patients at a large teaching hospital in Turkey (344 PTCs and 116 healthy controls).	VDD and insulin resistance more likely in PTC patients.	Log-VitD3 regression analysis exhibited significance with respect to tumor diameter.
Case- control	Kim et al. (2014)	548	South Korean females with PTC.	When separated into quartiles, those with VDD were significantly more likely to present with T3/4 tumors, LNM, and ETE.	Those with VDD were more likely to present with negative clinicopathologic features such as T3/4 disease, LNM, and ETE.
Case- control	Ahn et al. (2016)	820	Patients with PTC at a large university hospital in South Korea.	While 97% of the patients had insufficient vitamin D levels, no significance was found with respect to vitamin D and LNM, ETE, risk of recurrence, or advanced stage.	Findings were statistically insignificant.
Case- control	Choi et al. (2017)	560	“The Cancer Genome Atlas” consortium of data. 501 PTCs and 59 healthy controls.	VDR mRNA was upregulated in PTC tissues when compared to their healthy counterparts.	High VDR mRNA expression was significantly associated with T4, N1b, and AJCC/MACIS stage 4 PTC.
Case- control	Warakomski et al. (2018)	177	Patients with newly diagnosed PTC at a Polish cancer center.	No significant findings between 25-hydroxyvitamin D and tumor size were observed.	Findings were statistically insignificant.
Case- control	Beysel et al. (2018)	337	165 PTCs and 172 healthy controls from a large research hospital in Turkey.	Varying Fokl CDR polymorphisms may contribute to PTC risk (OR varying from 1.71–2.44 depending on polymorphism).	Fokl gene polymorphisms were significantly associated with later stage and negative prognostic factors depending on the genotype. Fokl genotype could be a useful prognostic or risk factor for PTC.
Case- control	Roehlen et al. (2018)	859	257 DTC, 139 HT, and 463 NC via a multicenter study in Frankfurt, Germany.	1α,25-hydroxyvitamin D interacts with FOXp3 as a mediator exerting its downstream effects. Potential therapeutics for DTC could target vitamin D as well as SIRT activators.	N/A
Case- control	Sulibhavi et al. (2019)	334	Patients with PTC at an urban, safety net hospital.	In those with a history of VDD, 13.9% were more likely to have advanced PTC.	Findings were statistically insignificant.
Cohort	Roskies et al. (2012)	212	Patients undergoing thyroidectomy for suspicious nodules at a university affiliated thyroid cancer center.	Low 25-hydroxyvitamin D levels conferred a RR of 2.0 (95% CI: 1.07–2.66) for well-differentiated thyroid cancer upon final pathology report.	N/A
Cohort	Danilovic et al. (2016)	433	Patients undergoing thyroidectomy for thyroid nodules. 187 patients were ultimately diagnosed with PTC.	No significant conclusion drawn between 25-hydroxyvitamin D serum levels and PTC risk or negative prognostic factors.	Findings were statistically insignificant.
Cohort	Jeong et al. (2020)	279	154 pHPT and 125 sHPT patients undergoing parathyroidectomy via a multicenter study in South Korea.	Serum 25-hydroxyvitamin D levels were lower in those with PTC and HPT when compared to HPT alone. In addition, 9.1% of pHPT and 7.2% of sHPT cases were concomitantly diagnosed with PTC.	Those with concomitant disease (PTC and HPT) were more likely to have microcarcinomas with aggressive features.
Cohort	Abdellateif et al. (2020)	190	70 PTC, 60 benign thyroid nodules, and 60 normal controls.	There was a significant decrease in the serum level of 25-hydroxyvitamin D in those patients with PTC when compared to NC and BN. PDGF and IGF-1 serum levels showed a significant inverse relationship when comparing the PTC and BN groups to NC.	N/A
Cross- sectional	Choi et al. (2017)	5186	Euthyroid patients who were evaluated with routine health checks at a large South Koreaninstitution. 53 patients were ultimately diagnosed with thyroid cancer.	Comparing mean levels of 25-hydroxyvitamin D of healthy patients and thyroid cancer patients proved insignificant. Among vitamin D subgroups, no significant difference of thyroid cancer prevalence was appreciated.	N/A
Case- control and meta- analysis	Hu et al. (2018)	276 participants,11 total studies	Chinese Han patients without cancer.	Case control: A significant inverse relationship between 25-hydroxyvitamin D serum levels and PTC risk (adjusted OR = 0.25). Meta-analysis: Pooled OR of thyroid cancer in vitamin D deficient patients was 1.42.	N/A
Meta- analysis	Zhao et al. (2019)	14 studies	N/A	Six of the studies showed a significant pooled OR of 1.30 with respect to low 25-hydroxyvitamin D and TC.	N/A
Literature review	Kim D. (2017)	N/A	N/A	Vitamin D has an ambiguous relationship with autoimmune thyroid disease and thyroid cancer. Further clinical trials are needed to determine the risk conferred by low vitamin D and the possible efficacy of vitamin D treatment.	N/A
In Vitro—Human Cells					
In vitro	Pang et al. (2020)	N/A	N/A	Infection of in vitro cell lines with VDR-enhanced lentiviruses showed pro-apoptotic, anti-proliferative, and anti-invasive characteristics. This supports the possibility for vitamin D to be used as a potential therapeutic agent.	N/A

VDD: vitamin D deficiency; LNM: lymph node metastasis; ETE: extrathyroidal extension; BN: benign nodules; NC: normal controls; TC: thyroid cancer; DTC: differentiated thyroid carcinoma; FTC: follicular thyroid carcinoma; ATC: anaplastic thyroid carcinoma; PTC: papillary thyroid carcinoma; VDR: vitamin D receptor; p/s HPT: primary/secondary hyperparathyroidism; HT: hashimoto’s thyroiditis; CI: confidence interval; OR: odds ratio; RR: relative risk.

## Data Availability

No new data were created or analyzed in this study, data sharing is not applicable to this article.

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
