# Peer review of "The Role of Vitamin D as a Prognostic Marker in Papillary Thyroid Cancer"

_cancers, 2021, doi:10.3390/cancers13143516_

Round 1

Reviewer 1 Report

Bains et al. reviewed scientific literature about the potential role of vitamin D in PTC development and aggressiveness. The review could be interesting since the role of vitamin D has been described in other cancers (although non consensus till now exists). However, the review needs significant improvement in order to increase its scientific interest. Overall, selected studies are merely described without any comment and discussion: revision of the literature should be analyzed and commented.

  • Introduction section, line#3: the reported thyroid cancer 2021 death rate estimation regards all histotypes and, in this context may be misleading. Rather, PTC death annual rate estimation should be reported.
  • Introduction section, page#2, lines#1-3: Again, not all thyroid cancer  types have the same good prognosis: PTC prognosis is excellent except for specific sub-types such as "tall cells", "insular" and so on.
  • Method of scientific literature searching and selection should be described, including the studies' language selected.
  • Table: The table should be modified to becoming easier to be read by grouping the reported studies according to study design and type (in vitro, in humans or animals).
  • Page#3: "patients with PTC had significantly lower serum vitamin D and estimated a sensitivity and specificity of 65.7% and 58.3%, respectively". PPV and NPV are very low and poorly useful in predicting PTC but, also in this case, no comment or discussion is reported by the authors.

Author Response

Bains et al. reviewed scientific literature about the potential role of vitamin D in PTC development and aggressiveness. The review could be interesting since the role of vitamin D has been described in other cancers (although non consensus till now exists). However, the review needs significant improvement in order to increase its scientific interest. Overall, selected studies are merely described without any comment and discussion: revision of the literature should be analyzed and commented.

-Introduction section, line#3: the reported thyroid cancer 2021 death rate estimation regards all histotypes and, in this context may be misleading. Rather, PTC death annual rate estimation should be reported.

Thank you for the valuable perspective. As these estimations were reported for all histologic sub-types of thyroid cancer, the delineation of the papillary sub-group was omitted from the original source. Therefore, extrapolation of known PTC incidence and mortality was required to report the estimated annual death rate. This is reflected in the change made to the manuscript text.

-Introduction section, page#2, lines#1-3: Again, not all thyroid cancer types have the same good prognosis: PTC prognosis is excellent except for specific sub-types such as "tall cells", "insular" and so on.

We have changed the manuscript text in order to more accurately reflect the vague nature of the reported thyroid cancer overall survival. Thank you for your feedback.

-Method of scientific literature searching and selection should be described, including the studies' language selected.

The study’s design has been expanded upon within the introduction in congruence with this feedback. Thank you for ensuring the study design remains transparent for the future readers.

-Table: The table should be modified to becoming easier to be read by grouping the reported studies according to study design and type (in vitro, in humans or animals).

Table 1 has been organized in an attempt to improve readability. Abbreviations were also added in the last row.

-Page#3: "patients with PTC had significantly lower serum vitamin D and estimated a sensitivity and specificity of 65.7% and 58.3%, respectively". PPV and NPV are very low and poorly useful in predicting PTC but, also in this case, no comment or discussion is reported by the authors.

Additional elaboration on this subject has been added during the discussion of Abdellateif et al.’s results.

Reviewer 2 Report

- Interesting and relevant topic.

  • Although this is not  a systematic review and just a simple literature review, It would be ideal if authors can elaborate somewhat of the search criteria they used to decide which students to include in this paper.
  • Conclussions mention:"While research regarding the use of vitamin D as a prognostic factor in papillary thyroid cancer remains controversial, several studies have found significant correlations be-tween vitamin D deficiency and the occurrence of thyroid cancer.  "

Several students on the subject (mentioned in this review) also failed to find this association (actually the largest studies). For this reason I believe the conclusion  (and certain other parts in which this topic is discussed in the introduction and abstract) should be modified to reflect that in fact the correlation between vit D levels and thyroid cancer is also still controversial as we have studies with conflicting results.

Author Response

Interesting and relevant topic.

-Although this is not  a systematic review and just a simple literature review, It would be ideal if authors can elaborate somewhat of the search criteria they used to decide which students to include in this paper.

The study’s design has been expanded upon within the introduction in congruence with this feedback. Thank you for ensuring the study design remains transparent for the future readers.

-Conclusions mention: “While research regarding the use of vitamin D as a prognostic factor in papillary thyroid cancer remains controversial, several studies have found significant correlations be-tween vitamin D deficiency and the occurrence of thyroid cancer.  "

Several students on the subject (mentioned in this review) also failed to find this association (actually the largest studies). For this reason I believe the conclusion  (and certain other parts in which this topic is discussed in the introduction and abstract) should be modified to reflect that in fact the correlation between vit D levels and thyroid cancer is also still controversial as we have studies with conflicting results.

Thank you for this valuable perspective. We have heeded your advice and changed the conclusion and discussion revolving around PTC and vitamin D in an attempt to make the controversial nature of these relationships more apparent.

Reviewer 3 Report

The authors present a review of the literature on the possible role of vitamin D in thyroid cancers. The topic is interesting and compared to other cancers that are still poorly studied with results that are still controversial. The authors' merit is to focus attention on this particular type of tumors and this is certainly positive;

in my opinion, however, the authors should consider the literature data regarding the potential effects of vitamin D on the prevention of bone metastases of PTC; in any case, I believe it would be appropriate, in addressing the relationship between vitamin D and thyroid cancer, to comment on the implications relating to the possible complications peculiar to surgical therapy of thyroid cancer in terms of potential alterations in bone metabolism

Author Response

The authors present a review of the literature on the possible role of vitamin D in thyroid cancers. The topic is interesting and compared to other cancers that are still poorly studied with results that are still controversial. The authors' merit is to focus attention on this particular type of tumors and this is certainly positive;

-In my opinion, however, the authors should consider the literature data regarding the potential effects of vitamin D on the prevention of bone metastases of PTC; in any case, I believe it would be appropriate, in addressing the relationship between vitamin D and thyroid cancer, to comment on the implications relating to the possible complications peculiar to surgical therapy of thyroid cancer in terms of potential alterations in bone metabolism

Thank you for bringing this aspect of our discussion into light. We have attempted to further elaborate on vitamin D and its role on metastases with respect to papillary thyroid cancer. In addition, a discussion revolving around thyroidectomy and bone metabolism have been added. Unfortunately, we were unable to find any studies examining vitamin D’s role in PTC’s ability to metastasize to bone in particular.

Reviewer 4 Report

This article is a brief but interesting review on the prognostic role of vitamin D as a prognostic marker for papillary thyroid carcinoma. In my opinion, however, a few minor changes need to be made before being accepted for publication, as follows:

  • The authors should specify the search criteria when reviewing the literature on this topic.
  • In the introduction, 2nd paragraph, line 7: Established molecular prognostic factors such as TERT promoter mutations and some related reference (Soares P, Póvoa AA, Melo M, Vinagre J, Máximo V, Eloy C, Cameselle-Teijeiro JM, Sobrinho-Simões M. Molecular Pathology of Non-familial Follicular Epithelial-Derived Thyroid Cancer in Adults: From RAS/BRAF-like Tumor Designations to Molecular Risk Stratification. Endocr Pathol. 2021 Mar;32(1):44-62. doi: 10.1007/s12022-021-09666-1. Epub 2021 Mar 2. PMID: 33651322.) should also be included.
  • In the introduction, third paragraph, line 4. Follicular-derived thyroid cancer is categorized in 3 major types: well-differentiated carcinoma, poorly differentiated carcinoma and anaplastic (undifferentiated) carcinoma. Medullary thyroid carcinoma is derived from C cells. Well-differentiated carcinoma includes papillary thyroid carcinoma, follicular thyroid carcinoma and oncocytic (Hürthle) cell carcinoma. As a consequence,  poorly differentiated thyroid carcinoma is not a differentiated thyroid carcinoma; this must be corrected.
  • The abbreviations from Table 1 should appear as a legend on the last line of Table 1.
